# Evaluation of Intervention Effectiveness of Sensory Compensatory Training with Tactile Discrimination Feedback on Sensorimotor Dysfunction of the Hand after Stroke

**DOI:** 10.3390/brainsci11101314

**Published:** 2021-10-02

**Authors:** Ken Kitai, Masashi Odagiri, Ryosuke Yamauchi, Takayuki Kodama

**Affiliations:** 1Department of Rehabilitation, Maizuru Red Cross Hospital, Kyoto 624-0906, Japan; fbgk0311@yahoo.co.jp; 2Department of Physical Therapy, Faculty of Health Sciences, Kyoto Tachibana University, Kyoto 607-8175, Japan; odagiri@tachibana-u.ac.jp (M.O.); yamaryou2006@yahoo.co.jp (R.Y.)

**Keywords:** sensorimotor disorder, sensory compensation, sense of agency, tactile discriminative feedback

## Abstract

We investigated the intervention effect of training using a feedback-type tactile discrimination system on sensorimotor dysfunction of the hand after a stroke. A human male subject with sensorimotor dysfunction in his left hand after a stroke was asked to perform peg manipulation practice, a building block stacking task, and a material identification task for 10 min each for six weeks. During the activities, a tactile discrimination feedback system was used. The system is a device that detects the vibration information generated when touching an object with a hand and that feeds back the captured information in real time as vibration information. After the intervention, in addition to the reorganization of the sensorimotor areas, the deep sensation, sense of agency, numbness, amount of use, and quality of the left-hand movement improved. Our results suggest that training with the use of a feedback system could be a new form of rehabilitation for sensorimotor dysfunction of the hand.

## 1. Introduction

Approximately 85% of stroke patients experience hemiplegia as an aftereffect. One of the most common symptoms of hemiplegia is sensory dysfunction of the hand, which is closely related to upper limb motor function [1]. Among hemiplegic patients, 60% are unable to use the function of the affected hand in daily life and have impaired activities of daily living (ADL) [2,3,4]. The sensory function of the hand plays a role in detecting small fluctuations or errors in muscle contraction during motor execution as well as in reflexively regulating movement [5,6]. When feedback from the somatosensory system is blocked or attenuated in the different pathways of the central nervous system due to some effect, the muscles can adjust reflexively via somatosensory sensation; when grasping an object with a hand, the central nervous system shows an excessive output of grasping force, and the variation of force during the movement also becomes large, resulting in the object being dropped [7]. This is called sensorimotor dysfunction of the hand and can appear of which afferent pathway of the central nervous system is damaged—from the posterior spinal cord to the brain cortex [1,8]. Sensorimotor dysfunction of the hand is thought to prevent the recovery of motor function in the affected hand [9]. To evaluate the approaches that deliver feedback from sensory stimuli, visual [10,11,12], electrical [13,14], and auditory [15] stimuli have been used for the rehabilitation of sensorimotor dysfunction in hemiplegic patients.

As a result of interventions using these methods, some effects have been reported, including functional reorganization of sensorimotor areas in the brain and refinement of coarse motor functions [10,11,12,13,14,15]. The effectiveness of these methods may be associated with the potential of the brain to process information synchronously between motor intent and sensory feedback of the treated limb. Synchronous information processing means that motor intentions and sensory feedback are processed in the brain within 250 ms without any gaps, generating a “motor subjective feeling” that allows the subject to perform the movement [16]. This reportedly increases the activity of the premotor cortex and corticospinal tract [17,18]. The use of electroencephalography (EEG) has enabled the detection and evaluation of brain activity at rest and during movement, advancing the exploration of these physiological processes [19].

In the ipsilateral hand, which has precise functional characteristics, feedback from visual information is effective at capturing the target object, but it lacks sensory information to track the information inside the body. This results in overt motor output of the hand [20], making it difficult to control the friction that occurs between the fingers and the object (i.e., “kinetic friction”) with the muscle activity of the hand. Electrical stimulation and auditory stimulation are also difficult to detect in terms of both kinetic friction and continuous feedback [13,14,15]; therefore, they may be inadequate for measuring feedback stimuli on the hand, which is required to precisely move according to the shape of the object. Somatosensory information on the hand’s muscle activity is needed to predictively use visual information to confirm the coordinates of the target object and control the dynamic friction generated between the object and finger muscle belly [20]. To do this, it is necessary to maintain both sensory inputs in a synchronized order. However, to date, no approach has been developed to satisfy these conditions.

To develop a new method for rehabilitating sensorimotor dysfunction of the hand, we focused on an approach using a deep sensory vibrating stimulus that compensates for the frictional information input received by the hand. As deep sensation can capture the pressure changes associated with articulation movement, it can be used as a compensatory stimulus to control the movement of the hand. It has been reported that the more severe the impairment of a deep sensory finger function, the more the person is unable to maintain a certain muscle force overtime and the more the person drops objects held by the affected hand [21]. This means that compensating for the constantly changing tactile information from the object and finger muscle belly is important for reorganizing elaborate movements in patients with sensorimotor dysfunction of the hand.

The purpose of this study was to validate the usefulness of a compensation approach strategy by conducting hand rehabilitation while using a Yubi-Recorder, a tactile discrimination feedback device that provides compensation and synchronous feedback of constantly changing tactile sensory information to the hand through vibration stimulation.

## 2. Materials and Methods

### 2.1. Case Introduction

A 52-year-old right-handed man who developed a right capsular hemorrhage approximately four years ago underwent hemoperfusion at Hospital A (Figure 1A), developed a right coronary infarction approximately one year later (Figure 1B), and was admitted to Hospital B. These factors resulted in paralysis of the left upper and lower limbs. The right upper and lower limbs were at a level that did not interfere with daily life. He availed the home rehabilitation service of Hospital B once a week. For this, the physical therapist gave him 20 min of left upper and lower limb joint range of motion training, muscle strengthening training, and gait training to maintain his functional ability. Approximately four years after the right capsular hemorrhage, he fell at home and was brought to Hospital B. Computed tomography (CT) showed that he had a plateau fracture of the left tibial outer condyle and a fissure fracture of the intercondylar comb. The patient was admitted to a regional hospital ward. The patient was treated for four weeks with hybrid seine fixations and underwent physical therapy with a 1/3 load increase every week. As part of his physiotherapy, they assessed his left upper extremities and found that the numbness on his left-hand side was extremely strong (score: 10) on the numerical rating scale (NRS) (0: not felt at all, 10: felt extremely strongly). He also had a loss of somatosensory and thermal pain sensation in his left upper and lower limbs. In terms of motor paralysis, the patient’s Brunnstrom stage was III in the left upper extremity and IV in the left fingers. In the left-hand elaboration task, the patient was asked to show his ability to carry small objects, but controlling the manipulation of objects was difficult because of ataxia symptoms affecting his hand. The patient reported that he could not feel his hands, so sensorimotor dysfunction of his left hand was suspected. About one month after admission, the patient was transferred to a recovery ward at Hospital B for continued rehabilitation through physical therapy. In addition, occupational therapy for sensorimotor dysfunction of the left hand was initiated after his stay in the recovery ward, and joint range of movement training, muscle strengthening, and electrical stimulation of the left hand were added to the approach. In addition, a motor task using the Yubi-Recorder for sensorimotor dysfunction of the left hand was performed five times per week for 30 min each time for six weeks. We confirmed that the patient had motor functions that allowed him to perform the research tasks prior to taking measurements. We also confirmed that there were no problems with cognitive function, communication, or any abrupt seizures (i.e., poor general state), such as epilepsy. This study was conducted in accordance with the guidelines of the Declaration of Helsinki, and the study was approved by the Kyoto Tachibana University Ethical Review Committee (Approval No. 20-33). The purpose, content, and procedures of the study were fully explained to the participants orally and in writing, and their consent was obtained.

### 2.2. Initial Physical Therapy Assessment

With respect to sense ratings [22], the position sense was 5/5 correct on the left hand. For motor perception, there were only 2/5 correct answers on the left hand, indicating deep sensory loss. On the left-hand side, the numbness was 10/10 on the NRS, and a strong numbness remained. On the Pain Catastrophizing Scale (PCS) [23], which rates the cognitive aspect of numbness, the cutoff value was set at 29/52 [24]. The patient was ruminating at 19/20, helpless at 15/20, and magnified at 9/12, which resulted in a total score of 43/52, suggesting catastrophic thinking.

The Fugl-Meyer Assessment (FMA), a comprehensive assessment of motor function [25,26], showed that the patient had a lower upper extremity motor score of 25/66. FMA scores in the range of 23–31 points indicate that a patient does not have an upper extremity ability to perform ADLs [25] and compensates by using the right upper extremity for performing most activities. His simple test for evaluating hand function (STEF) [27], which quantitatively evaluates the ability to perform subtle hand movements, showed a score of 97/100 on the right hand and 5/100 on the left hand; he also scored low for the ability to carry objects. A Purdue pegboard test (peg test) [28], which measures rough hand movements and node manipulations, showed a median value of 1.5 ± 0.5 on the left-hand side.

In the learnability assessment, we heard statements that were indications of avoidance behavior and decreased awareness of one’s own body, including “I don’t use my left hand at all”, providing evidence that the sense of agency and sense of body ownership of the patient were decreased. Thus, using the NRS, we asked, “How much do you feel that you are the one who moves your own hand?” and “How much do you feel your numb hand is your own body?” The elicited sense of agency and sense of body ownership of the left hand decreased at 2/10 and 3/10, respectively, indicating a decreased awareness of one’s own body [29]. The Japanese version of the motor activity log (MAL) [30,31], which is used to evaluate the use of the hand related to ADL and quality of life (QOL), is a questionnaire that consists of items for amount of use (AOU) and quality of motion (QOM) of the hand. The answers were scored on a 6-point scale (0 = no use, 5 = normal). The results showed that the AOU value was 3/56, and the QOM value was 4/56. The left-hand side was rarely used in daily living, as reported by the patient, and the quality of motion was also low. In addition, an NRS question for sense of agency was asked for each item of the MAL, and it was confirmed that the sense of agency was decreased at 4/140.

### 2.3. Intervention and Evaluation

The hand is a tactile receptor that captures the kinetic friction generated by manipulating an object and controls the subtle muscles in response to friction. Considering these, a system device capable of detecting and feeding back the kinetic friction generated when touching an object with a hand is necessary to compensate for the function of a sensitive tactile discrimination site such as the hand. To address these issues, Tanaka et al. [32] developed the Yubi-Recorder, a system device that detects vibration information generated when touching an object, captures minute changes in friction generated on a fingertip, and feeds back the captured information in real time as vibration information. Essentially, the Yubi-Recorder (Tech Giken Co., Ltd., Kyoto, Japan) is a device that can measure vibration information by detecting any vibration of the skin that occurs when an object is touched. More specifically, a Yubi-Recorder can sense unequalness, flatness, curvature, and rudeness, and it can capture tactile stimuli from an object of any shape, adequately handling multi-directional motion. The sensor that can detect vibration is attached to the distal interphalangeal (DIP) joint of the index finger, and vibration information can be presented through an oscillator by modulating the sensor information to the frequency felt by humans. Regarding the instruction on intervention, we explained, “When something touches your hand, the terminal on your shoulder vibrates in real time. In addition, the intensity of the vibration varies depending on the feeling of the object. Feel the difference in these vibrations and connect the difference in your head to the material you feel at your fingertips.”

The approach comprised (1) a 10 min task of stacking square blocks with a base length of 3 cm used in a course cube test with the left hand, (2) a 10 min task of discriminating between five sandpapers using the ventral side of the left index finger, and (3) a 10 min task of placing a 25 mm long, 3 mm diameter iron peg (Sakai Medical) on a board with two vertical lines. The task of inserting the iron peg into a board with two vertical lines of 25 holes (peg manipulation practice) was performed with the left-hand side. A Yubi-Recorder was attached to the DIP joint of the index finger, and tactile information of the ventral skin of the index finger was detected by the tactile sensor as vibrations. Tactile stimuli were synchronously presented to the body through vibrations; the assembly site of the vibrator was the left acromion or the left temporal bone. Sense discrimination [33] has been used in previous studies; considering this, five types of sandpaper with different roughness values were applied to the left acromion and right temporal skeleton. These were judged to be the areas where the roughness of the sandpaper could be best identified. The intervention period lasted six weeks, and the approach was performed five times per week.

The participant first performed the peg manipulation practice without the Yubi-Recorder attached to his left hand (no Yubi-Recorder condition). The participant then performed the peg manipulation practice with a Yubi-Recorder attached to his left hand (finger recorder condition).

To verify the effect of hand rehabilitation with the Yubi-Recorder, we performed pre- and post-evaluations from the perspective of sensory feature evaluation of the left hand, motor feature evaluation of the left upper limb, learnability evaluation, and EEG after six weeks of intervention (Evaluation A). In addition, a peg test was performed at the end of each week to evaluate changes in the motor function of the left hand over time (Evaluation B).

Deep sensory tests of the hand, including positional and motor sensations, have been performed to evaluate sensory features such as NRS and PCS [23] and to evaluate the degree of numbness. The deep sensory tests performed in this study were positional and kinesthetic tests. In the positional test, the examiner moved the subject’s hand and asked the subject to mimic the movement with the opposite hand. In the kinesthetic test, the subject was first asked to close his eyes; the examiner then moved the affected side of the mother finger in an altruistic manner, and the subject was asked to verbally identify the direction in which the finger was moved. For the 14-item PCS questionnaire, all references to pain were converted to numbness, and the subjects were asked to identify their degree of numbness on a 5-point scale (0 = not true at all, 4 = very true). The participants were asked to complete the modified test to assess the following: state of being unable to get the numbness out of one’s mind (rumination), feeling helpless to do anything about numbness (helplessness), and thinking that the numbness will be stronger than it actually is (magnified vision).

Motor function evaluations of the top extremities included the FMA [26], STEF [27], and peg test [28]. The FMA has test items for shoulder/elbow/forearm, wrist, fingers, and coordination/speed, and it evaluates the overall motor function of the upper limb, including single and compound articulations of the upper limb, grasping of objects, coordinated movements of the upper limb, and motor speed of the upper limb. STEF consists of 10 trials, with each trial consisting of grasping and moving an object of varying size and shape, measuring the time required to accomplish a series of movements with a stopwatch, and adding 1–10 points depending on the time it took to perform a task.

The peg test involves inserting as many iron pivots as possible into a board with two lines of 25 holes within 30 s. Grasping force refers to the time and acceleration measured when the grasping force is applied to a grasping force gauge (Tech Giken Co., Ltd., Kyoto, Japan). This device was used to measure the vibration generated in three axes (X, Y, and Z). In order to unify the distances traveled by vibration in these axes, two pieces of paper (53.7 cm width, 25.4 cm length) were placed side by side to perform a box and block test [34]. This manual evaluation is commonly used after a stroke. A marker was placed at the midpoint of the left and right papers separated by a partition, and a grasping force gauge was then set to move from the left to the right midpoint. Afterward, 5 g, 10 g, 17.8 g, 31.2 g, and 62.4 g weights were individually attached to the grasping force gauge and were evaluated before being loaded. To evaluate the time, the mean value and standard deviation of the recorded triples were obtained. The average time required to accomplish the grasping force task and the average value of the three-axis composite values were then calculated. The rapid movement of a grasping force gauge requires this task to decrease the task execution time. Thus, it is necessary to move the upper limbs quickly to manipulate the grasping force gauge. However, when the grasping force gauge is moved quickly, the acceleration in the grasping force gauge and the hand increases, making the three-axis composite value larger. Hence, when performing the task quickly, the acceleration of the finger belly needs to be controlled, so the hand does not drop the grasping force gauge. Thus, the faster the average time it takes to accomplish the grasping force task, the more fluent the upper limb is moved while controlling the acceleration generated by the grasping force gauge and the hand within the fingertip.

To assess learnability, we used the NRS to score the sense of agency and body possession in the same manner as the MAL [30,31] was used for ADL and QOL of the hand. An NRS on sense of agency [29] was provided for each MAL questionnaire item, and the relationship between behavior change and sense of agency was also rated. EEG was conducted for neurophysiological evaluation. EEG was measured using active electrodes (G.TEC Medical Engineering, Schiedlberg, Austria) and a bio-signal measurement device (Livo, Tech-Gihan, Kyoto, Japan). Because there is evidence that the degree of motor learning can be predicted by comparing and verifying sensorimotor features at rest and during exercise using EEG [35], we recorded results during one minute of rest and during certain tasks. Specifically, EEG was measured while the participant performed the peg manipulation practice with his left hand with and without the Yubi-Recorder. Using both earlobes as references, the EEG measurement sites were derived from 15 sites (Fpz, Fz, Cz, Pz, Oz, F3, F4, C3, C4, P3, P4, F7, F8, T7, and T8) based on the international 10–20 system. Previous studies have shown that 15 channels are sufficient to detect sensorimotor area activities and, in particular, frontal and parietal lobe activities associated with sensorimotor area information processing [36]. The bandpass filter was set at 0.5–60 Hz, and the sampling rate was set at 1000 Hz. The data measured by EEG equipment were spatially analyzed with accurate low-resolution brain electromagnetic tomography (eLORETA) analysis, which is a three-dimensional imaging method for neural activity in the brain [37]. For EEG data, we computed the coordinates in the x, y, and z directions in a brain region divided into 6239 voxels by a linear transformation, and then we transformed it into a three-dimensional image via nonlinear transformation with curved anatomical correction. As a result, the neural activity area for each task condition was computed as the value of neural activity (μV/mm^2^) at each voxel and was expressed as the Brodmann area (BA). In addition, to compare the immediacy of the Yubi-Recorder and the results before and after the intervention, we used the eLORETA statistical nonparametric mapping toolbox 26 (2007 ver.), a multiple paired t-test with nonparametric randomization [38].

The study protocol consisted of a 1-week pre-intervention assessment (Assessment A), which was followed by a 6-week intervention period and a longitudinal assessment (Assessment B). A post-intervention evaluation (Evaluation A) was conducted for a week to complete the study.

## 3. Results

After sensory feature evaluation, the motor perception of the left hand improved from 2/5 to 5/5. Numbness on the left-hand side also improved, decreasing from NRS 10 /10 to 6/10. Additionally, PCS improved with the following changes: 19/20 to 15/20 for rumination, 15/20 to 15/20 for helplessness, 9/12 to 6/12 for magnification, and 43/52 to 36/52 for total—exhibiting a drop in scores for rumination and magnification and an improvement in catastrophic thinking (Table 1). In terms of motor function, FMA had the following results: 11/36 to 23/36 for shoulder/elbow/forearm, 2/10 to 3/10 for wrist, 12/14 to 10/14 for fingers, 0/6 to 0/6 for coordination/speed, and 25/66 to 36/66 for the total score. FMA of the left upper extremity showed an improvement from 25/66 to 36/66; in STEF, the left-hand side showed a 5-fold improvement, increasing from 5/100 to 25/100 (Table 1). In the peg test, the median value of the left-hand side improved from 1.5 ± 0.5 to 2.0 ± 1.0 (Figure 2); the time to complete the task was reduced from 11.82 ± 1.85 to 6.39 ± 0.84 compared with the initial evaluation. The triaxial composite value increased from 9.73 ± 0.03 to 10.00 ± 0.02 when compared with the initial evaluation (Figure 3A,B). For learnability assessment, MAL improved from 3/56 to 14/56 for AOU and from 4/56 to 14/56 for QOM; in addition, MAL’s Subjectivity Engine NRS improved from 4/140 to 20/140 (Table 1).

Regarding brain function evaluation, without the Yubi-Recorder, neural activity was increased in the left orbitofrontal cortex, left dorsolateral anterior cingulate cortex, left dorsolateral prefrontal cortex, and left supplementary motor area. With the Yubi-Recorder, neural activity was increased in the left secondary visual cortex, both visual association areas, the two primary somatosensory cortices, the two superior parietal lobules, and the two inferior parietal lobules (angular gyrus and supramarginal gyrus) (Figure 4A,B and Table 2). Compared with the initial evaluation, the final evaluation showed enhanced neural activity in the left secondary visual cortex, both visual association areas, both primary somatosensory areas, both superior parietal lobules, both inferior parietal lobules (angular gyrus and supramarginal gyrus), and the right primary motor cortex (Figure 5, Table 2).

## 4. Discussion

This study investigated the usefulness of a feedback-type tactile discrimination system as a sensory compensatory training approach during rehabilitation in a post-stroke case with sensorimotor dysfunction of the left hand. More specifically, this was done using the Yubi-Recorder, a system device capable of providing compensatory and synchronous feedback of tactile sensory information on the hand through vibration stimulation.

In terms of sensory features, motor perception of the left hand improved from 2/5 to 5/5. It has been reported that when sensory feedback associated with movement is appropriately provided in real time, sensorimotor areas are reorganized, and deep sensory perception can be used to detect motor errors that occur between motor predictions and actual movements [39]. The same thing was thought to have occurred in this case. Thus, the idea was to illustrate the reorganization of the sensorimotor cortex based on EEG results; the use of a fingertip recorder allowed for the detection of motor errors that occurred between motor prediction and actual movement by providing sensory feedback in real time. Finally, the reorganization of the sensorimotor cortex allowed us to instantly detect motor errors caused by his own movement during the deep sensory examination of the left hand without using finger recollection. Furthermore, the numbness of the left-hand side improved from an NRS score of 10 to 6. This may be attributed to the ability of humans to block sensory information that is not necessary for locomotion [40]. Because the patient’s numbness improved after suppressing the sensory input—information not necessary for locomotion—the frictional information necessary for locomotion was input in real time or when the object was touched. Numbness is thought to be induced by a temporal mismatch between motor intentions and information matching on sensory feedback [41]. In this study, the sense of agency on the left-hand side was also determined, and the mismatch between sensation and movement may have contributed to the chronic numbness originally caused by organic changes in the afferent conduction pathways in the central nervous system due to stroke. The temporal coincidence of information matching between motor intent and vibration in the patient was also considered to have generated a sense of agency and to have improved numbness due to cognitive aspects.

The FMA score improved upon the motor feature evaluation. In the final evaluation, EEG results showed that the neural activity of the two superior parietal lobes, the two inferior parietal lobes (angular gyrus and supramarginal gyrus), and right primary motor cortex [42], which play important roles in forming motor imagery during left-hand acceleration, were enhanced. This might be due to the increased activity of the sensorimotor cortex in this region. On the other hand, the improvement was not recognized in the grasping power item or cooperation/speed using the finger. It seemed to be important to take the individual approach for motion paralysis and muscle force lowering because the Yubi-Recorder does not measure the force and motion speed of the arm and hand. In addition, both the STEF and peg tests improved, exhibiting improvements in motor function of the left hand, ability to carry objects, and elaboration; after measuring the grasping force of the left hand, the three-axis composite value was larger at the end than at the beginning. These observations might be explained by the fact that deep sensation captures the pressure changes associated with joint movement, thereby performing fine motor control of the hand [21].

The MAL results show that AOU and the quality of left-hand movement in daily life improved. Abnormal sensory perceptions, such as numbness, are known to cause learned disuse due to unpleasant emotions and to enhance interhemispheric inhibition from a healthy motor cortex [43]. For example, the patient reported, “My left hand is numb and I don’t like moving it”, indicating that he felt biased towards disuse of his left hand due to an emotional response that occurred before the intervention. In the final evaluation, numbness and PCS scores improved, suggesting that cognitive aspects of numbness and tingling improved, which in turn improved the amount and quality of use. Furthermore, behavioral changes are said to occur through motor learning [44], and the case was also thought to have improved AOU and quality of movement in daily life due to an improved sense of agency.

The left frontal eye region, left dorsal anterior cingulate cortex, dorsolateral left prefrontal cortex, and left supplementary motor region showed dominant neural activity during the intervention without a Yubi-Recorder. The orbitofrontal cortex is considered a region related to spatial attention [45]. This cortex is involved in determining the amount of control required to perform the intended movement [46]. The dorsolateral prefrontal cortex is involved in attention, memory, and cognition, and it is considered important in performing certain tasks [47]. The supplemental motor cortex is involved in motor planning and initiation of movement [48]. The results show that neural activity in the right front motor region increased when performing elaborate movements with the left hand, resulting in attention to movement, planning of movement necessary to perform the elaborate movements, and the controlling of elaborate movements being compensated. However, the compensatory enhancement of neural activity in the right motor cortex region on the healthy side of the patient is believed to inhibit functional recovery of the affected hand [49]. Furthermore, when the intended motion fails using an affected limb, the paralyzed limb becomes unused for fear of feeling failure and other emotions, which makes recovery of the motor function more difficult to achieve for the affected hand [43]. Moreover, in the case of the patient, his sense of agency was diminished, indicating that information matching between motor intention and sensory feedback was not temporally consistent; the patient learned that his intended moves failed, resulting in his left hand being disused. Against this background, right-hand motor functions were impaired despite more than three years since the onset of stroke. This might be due to an offsetting increase in neural activity of the right motor cortex area on the healthy side of the patient, as well as the possibility that recovery of the motor function of the left hand was inhibited by the learned disuse.

In contrast, in the condition with the Yubi-Recorder, the left secondary visual cortex, the two visual association areas, the two primary somatosensory cortices, the two superior parietal lobules, the two inferior parietal lobules (angular gyrus and supramarginal gyrus), and the two primary motor cortices showed dominant neural activity. The visual association cortex is a region related to spatial attention, which cues the direction of attention to a specific location and sends information acquired visually to the superior parietal lobule [50]. The fundamental somatosensory cortex is the site where somatosensory information from the periphery flows to the brain cortex in real time [51]. The superior parietal lobule integrates somatosensory information obtained from the skin with information obtained through spatial attention [52] to confirm the location of the body and the orientation of objects. Auditory and other information are added and stored as a variety of complex cognitive information in the inferior parietal lobule; information necessary for movement is then sent to the primary motor cortex to control the movement of the hands [53,54]. These findings suggest that compensatory sensory input measured by the Yubi-Recorder allowed sensory processing pathways in both parietal lobes to process information necessary for manipulating an object and to perform motor control of the left hand using the two primary motor areas.

To verify the effects before and after the intervention, we evaluated neural activity during left-hand peg manipulation at both the beginning and end stages, identifying their differing values. The results show that the left secondary visual cortex, both visual association areas, both primary somatosensory areas, both superior parietal lobules, both inferior parietal lobules (angular gyrus and supramarginal gyrus), and the right primary motor cortex were predominantly activated in the final evaluation. In this study, these results suggest that a six-week training period using the Yubi-Recorder has the potential to reconstruct a sensory information integration system centered on both parietal lobes and to improve motor control function when the left hand performs an acceleration move.

Despite these findings, this study has several limitations to consider. First, it is not possible to separate the effects of the sensory feedback device from those of the sustained exercises. Cortical activation patterns recorded by EEG are distinct from cognitive and motor stimulation associated with intensive interventions and cannot be attributed to the specific effects of new rehabilitation techniques. Second, we cannot establish that the clinical, psychological, and electrophysiological improvements were not due to the idiosyncrasies of one selected patient, and there is no guarantee that these will be seen in other patients. For comparative validation, interventions using the Yubi-Recorder should be performed in normal subjects and in those with motor and sensorimotor disorders of the hand. Finally, our results could not be analyzed for statistical significance. For this, we would like to increase the number of subjects, set up a control group, and perform a stratified randomized controlled trial to deeply examine the reliability of this rehabilitation approach.

## 5. Conclusions

We investigated the usefulness of a sensory compensatory strategy in this study by providing a rehabilitation approach that makes use of a tactile discrimination feedback system device, the Yubi-Recorder, for a patient who presented with sensorimotor dysfunction of the left hand after a stroke. Apart from the reorganization of sensorimotor areas, the approach resulted in improvements in depth of sensation, sense of agency, numbness, and both amount of use and quality of movement for the left hand. These results suggest that the rehabilitation approach that makes use of the Yubi-Recorder may be an effective rehabilitation method for patients with sensorimotor dysfunction.

## Figures and Tables

**Figure 1 brainsci-11-01314-f001:**
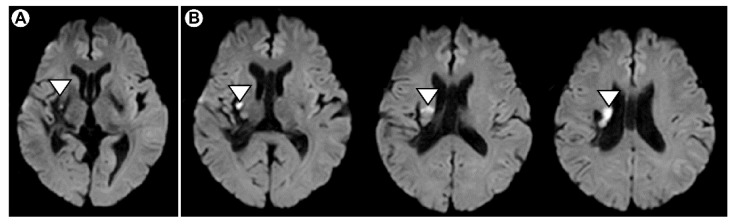
MRI at the time of diagnosis of right coronary cerebral infarction at Hospital B: (**A**) (pineal level) right capsular hemorrhage treated with hemoperfusion; (**B**) (lateral ventricular body level) high signal area in the right corona radiata.

**Figure 2 brainsci-11-01314-f002:**
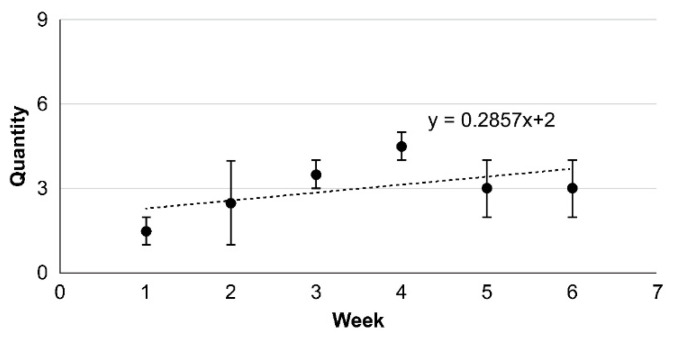
Peg test scores. In the initial evaluation, the median value was 1.5 ± 0.5. In the final evaluation, the median value was 3 ± 1, indicating improvement in elaborate left-hand movements.

**Figure 3 brainsci-11-01314-f003:**
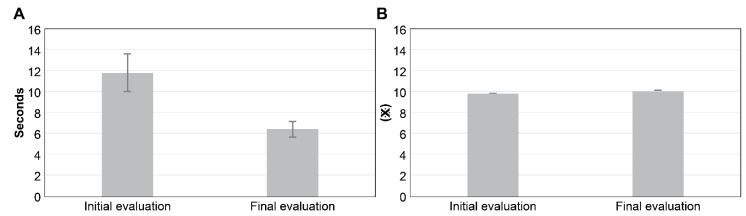
Grasping force measurements. (**A**) Time required to complete the grasping force task; (**B**) three-axis composite value. ※ Unit: m/s^2^. The time to complete the task was reduced from 11.82 ± 1.85 to 6.39 ± 0.84 compared with the initial evaluation. Compared with the initial evaluation, the triaxial composite value increased from 9.73 ± 0.03 to 10.00 ± 0.02.

**Figure 4 brainsci-11-01314-f004:**
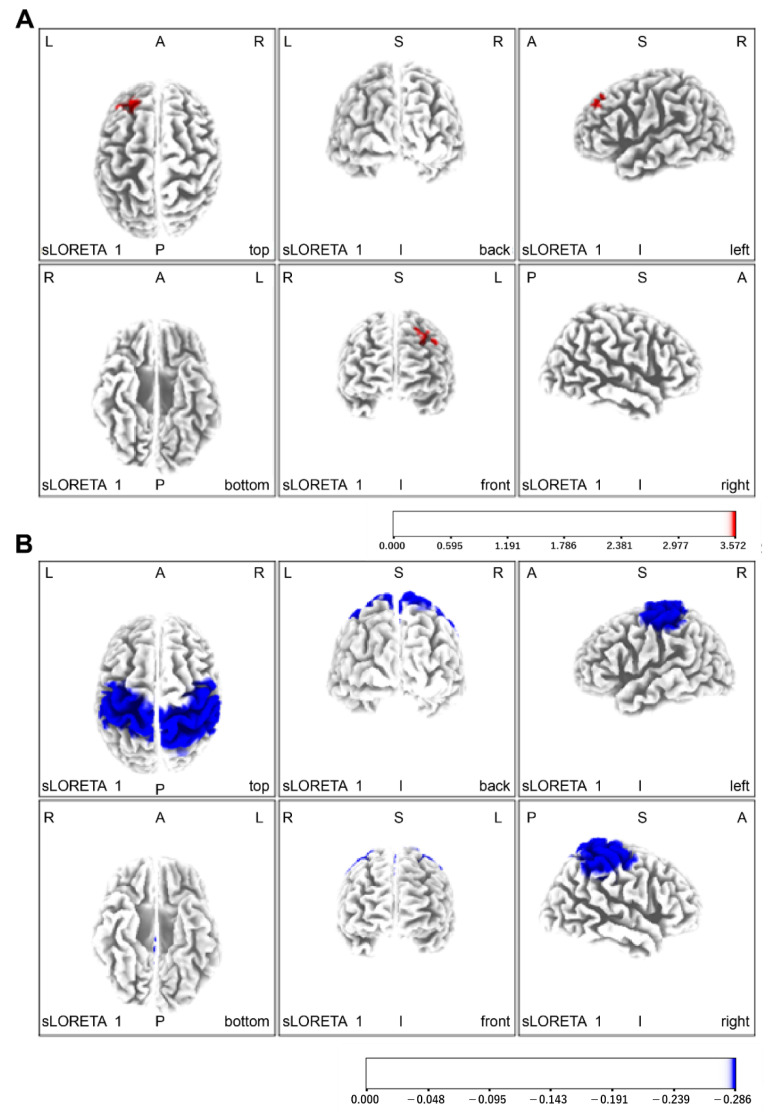
Areas of increased neural activity: (**A**) no Yubi-Recorder (red); (**B**) with Yubi-Recorder (blue) conditions. In the condition without the Yubi-Recorder, neural activity was increased in the left orbitofrontal cortex, left dorsolateral anterior cingulate cortex, left dorsolateral prefrontal cortex, and left supplementary motor area. In contrast, in the condition with the Yubi-Recorder, neural activity was increased in the left secondary visual cortex, both visual association areas, the two primary somatosensory cortices, the two superior parietal lobules, and the two inferior parietal lobules (angular gyrus and supramarginal gyrus).

**Figure 5 brainsci-11-01314-f005:**
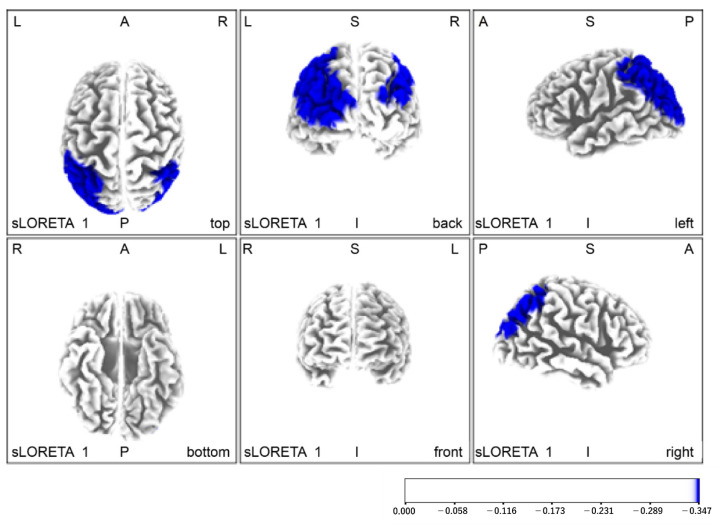
Areas of increased neural activity in the final assessment. Compared with the initial evaluation, the final evaluation showed enhanced neural activity in the left secondary visual cortex, both visual association areas, both primary somatosensory areas, both superior parietal lobules, both inferior parietal lobules (angular gyrus and supramarginal gyrus), and the right primary motor cortex.

**Table 1 brainsci-11-01314-t001:** Pre- and post-physical therapy assessment.

Assessment Description	Initial Evaluation	Final Evaluation
Position sense (times)	5/5	5/5
Motor perception (times)	2/5	5/5
Numbness (NRS)	10/10	6/10
PCS (Point)	Total: 43/52 Rumination: 19/20 Helplessness: 15/20 Magnification: 9/12	Total: 36/52 Rumination: 15/20 Helplessness: 15/20 Magnification: 6/20
FMA (Point)	Total: 25/66 Shoulder/elbow/forearm: 11/36 Wrist: 2/10 Fingers: 10/14 Coordination/speed: 0/6	Total: 36/66 Shoulder/elbow/forearm: 23/36 Wrist: 3/10 Fingers: 10/14 Coordination/speed: 0/6
STEF (Point)	Right: 97/100 Left: 5/100	Right: 96/100 Left: 25/100
MAL		
AOU (Point)	3/56	14/56
QOM (Point)	4/56	14/56
Sense of agency (NRS)	4/140	20/140

NRS, numerical rating scale; PCS, Pain Catastrophizing Scale; FMA, Fugl-Meyer Assessment; STEF, simple test for evaluating hand function; MAL, motor activity log; AOU, amount of use; QOM, quality of movement.

**Table 2 brainsci-11-01314-t002:** Most neural active sites and Montreal Neurological Institute (MNI) coordinates for each condition.

Task Conditions						Brodmann Area	Neural Activity Values
	x	y	z	Brain lobe			(μA/mm^2^)
Peg manipulation practice (left hand)							
Without Yubi-Recorder (red) vs. With Yubi-Recorder (blue)							
Red	−25	40	45	Left anterior cephalic lobe	Anterior cephalic eye field	8	0.13
Blue	−30	−85	40	Left posterior cephalic lobe	Visual field	19	6.75
Pre-intervention (red) < Post-intervention (blue)							
blue	−40	−55	60	Left parietal lobe	supramarginal gyrus	40	6.76

## Data Availability

The data that support the findings of this study are available from the corresponding author, T.K., upon reasonable request.

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
