# Peer review of "Evaluation of Intervention Effectiveness of Sensory Compensatory Training with Tactile Discrimination Feedback on Sensorimotor Dysfunction of the Hand after Stroke"

_brainsci, 2021, doi:10.3390/brainsci11101314_

Round 1

Reviewer 1 Report

Topic is interesting and provides (if validated) a good original approach for stroke rehabilitation

Major issue: it is a one case study;

  • does not allow a separation of the effects of the studied technique from the effects of sustained exercise (also implied by the protocol);
  • does not allow the specific attribution of the EEG/cortical activation patterns to the specific effect of the new rehabilitation approach as opposed to those of cognitive and motor stimulation associated with any intensive intervention
  • does not guarantee that improvements (both clinical, psychological and electrophysiological) are not due to particularities of the one chosen patient and would still be found in other patients
  • (in my opinion such notable benefit as that found by the authors might be attained mostly in patients with learned non use due to sensory loss and would not replicate in patients with proper pyramidal weakness)

Solution: expanding the subject number and inclusion of a control group that would engage in the same activities but without the Yumi device

Minor issues:

  • Fugl-Meyer results are shown only as total scores; more details might have allowed for a clearer representation of the type of deficit and the exact apects that improved;
  • For someone not familiar with the Yumi device a clearer presentation of the proprioceptive feedback would be useful (how does it deliver to the subject the informations regarding movement, presure, textures, etc )
  • use of nonvalidated/customized measurement tools
  • improve readability 
  • presentation of the extensive investigation/evaluation procedures and results might be hindering an easy understanding of the article; a separation of the clinical data from the EEG (in two separate communications) might prove useful.

Reviewer 2 Report

The authors present a case study of an individual who is experiencing sensorimotor dysfunction after stroke. They implement an intervention using a feedback-type tactile discrimination system, in combination with motor practice, for a duration of 6 weeks (10 min/day). Following the intervention, the authors present changes in a number of behavioural outcomes, as well as EEG. From this work, they suggest that this intervention warrants future study as an rehabilitation intervention to improve sensorimotor function post-stroke. The case study is very detailed and the authors present the information clearly. I have a few minor points below that might strengthen the paper. 

The introduction would benefit from a short discussion of some of the more detailed physiology that would underpin the utility of their suggested technique, specifically as it relates to the what they will go on to measure with EEG. As it is now, the reader wouldn’t know that EEG is going to be collected, and this could strengthen the argument as to why, after this study, the intervention warrants further investigation. The authors have included some of this in the discussion, but it would be nice to have it included in the introduction to set the study up. 

Similarly, the authors prep 15 EEG electrodes rather than a full cap. Presumably, this is because they have specific hypotheses as to the regions or networks that would be affected by the intervention. Introducing this and dedicating some time to explaining it to the reader would be useful. 

Additionally, they collect EEG both during the motor task and at rest. Are changes in resting EEG expected then? Explicitly stating details like that would improve clarity on the rationale for including these measures and conditions. 

Are the differences that the authors show in the results clinically meaningful?

Round 2

Reviewer 1 Report

Authors have adressed most of the issues I have raised in the initial review.  Although I still do not agree with some of their statements/assumptions, that may be due to my personal view of the facts and since those statements/assumptions are not incorrect in an absolute manner I think that readers should appreciate for themselves whether they are convincing or not.

Although the title was not changed and there was no extension of the study group, the abstract clearly states the "one subject" approach, as the discussion and conclusions also do.  

The general idea and approach are interesting. The results (in the one patient that tried the procedure) are clinically meaningful. The evaluation procedure is extensive (that might be a problem in a larger study).

This should be followed by a really convincing controlled/randomized study, hopefully warranting a future extended use of the procedure.

Minor comments - not warranting a new review:

  • at line 206 I still find the expression: "moved the affected side of the mother finger in an altruistic manner" somehow foggy
  • at line 344 there is an extra "the" in “by the his own movement”